# Dynamic Prediction of Resectability for Patients with Advanced Ovarian Cancer Undergoing Neo-Adjuvant Chemotherapy: Application of Joint Model for Longitudinal CA-125 Levels

**DOI:** 10.3390/cancers15010231

**Published:** 2022-12-30

**Authors:** Koceila Amroun, Raphael Chaltiel, Fabien Reyal, Reza Kianmanesh, Aude-Marie Savoye, Marine Perrier, Zoubir Djerada, Olivier Bouché

**Affiliations:** 1Department of Digestive and Endocrine Surgery, Université de Reims Champagne-Ardenne, VieFra, CHU Reims, 51100 Reims, France; 2Department of Medical Oncology, Godinot Cancer Institute, 51100 Reims, France; 3Department of Surgical Oncology, Godinot Cancer Institute, 51100 Reims, France; 4Department of Gastroenterology and Digestive Oncology, Université de Reims Champagne-Ardenne, Robert Debré Hospital, CHU Reims, 51100 Reims, France; 5Department of Pharmacology and Toxicology, Université de Reims Champagne-Ardenne, HERVI, CHU Reims, 51100 Reims, France

**Keywords:** ovarian neoplasms, CA-125 antigen, joint model, dynamic prediction, neoadjuvant therapy, cytoreduction surgical procedure, biomarker

## Abstract

**Simple Summary:**

Neoadjuvant chemotherapy is used in patients with initially unresectable advanced ovarian cancer (AOC) to reduce the disease bulk. The CA-125 level depends on tumor burden changes. A joint model (JM) is a statistical tool used for dynamic prediction during follow-up. A JM of longitudinal CA-125 was assessed as a reliable predictive model for overall and free disease survivals. We developed a dynamic and individual model to predict complete resectability of AOC using patients’ and tumor characteristics combined with kinetics of CA-125 during neo-adjuvant chemotherapy.

**Abstract:**

In patients with advanced ovarian cancer (AOC) receiving neoadjuvant chemotherapy (NAC), predicting the feasibility of complete interval cytoreductive surgery (ICRS) is helpful and may avoid unnecessary laparotomy. A joint model (JM) is a dynamic individual predictive model. The aim of this study was to develop a predictive JM combining CA-125 kinetics during NAC with patients’ and clinical factors to predict resectability after NAC in patients with AOC. A retrospective study included 77 patients with AOC treated with NAC. A linear mixed effect (LME) sub-model was used to describe the evolution of CA-125 during NAC considering factors influencing the biomarker levels. A Cox sub-model screened the covariates associated with resectability. The JM combined the LME sub-model with the Cox sub-model. Using the LME sub-model, we observed that CA-125 levels were influenced by the number of NAC cycles and the performance of paracentesis. In the Cox sub-model, complete resectability was associated with Performance Status (HR = 0.57, [0.34–0.95], *p* = 0.03) and the presence of peritoneal carcinomatosis in the epigastric region (HR = 0.39, [0.19–0.80], *p* = 0.01). The JM accuracy to predict complete ICRS was 88% [82–100] with a predictive error of 2.24% [0–2.32]. Using a JM of a longitudinal CA-125 level during NAC could be a reliable predictor of complete ICRS.

## 1. Introduction

At the time of diagnosis of ovarian cancer, the disease is at an advanced stage for 70% of the patients [1]. The standard therapeutic treatment for ovarian cancer is based on initial surgery to obtain a complete resection of all macroscopic tumors, and an adjuvant chemotherapy [2]. However, in some cases of advanced ovarian cancer (AOC), extensive tumor spread precludes initial complete cytoreduction surgery [1]. Therefore, the use of neoadjuvant chemotherapy (NAC) before interval cytoreductive surgery (ICRS) is indicated to reduce the disease bulk [3]. To avoid an unnecessary laparotomy, the tumor’s response after NAC is evaluated by a preoperative CT scan [4] and explorative laparoscopy [5]. However, these investigations present limitations. The clinical usefulness of a preoperative CT scan to predict resectability ranged widely between studies with a sensitivity of 19.2% to 100% and a specificity of 56.7% to 100%, whereas a PPV and a NPV ranged from, respectively, 46% to 100%, and 43.3% to 100% [6]. This is because the CT scan may miss small, disseminated tumors. Several studies have described the explorative laparoscopy as a reliable tool to identify patients suitable for complete CRS. However, its negative predictive value ranges from 69–96% [7,8,9,10], which corresponds to 4–31% of inappropriate laparotomies that do not result in complete cytoreduction due to the presence of definitively unresectable disease [7,8,11]. Indeed, during explorative laparoscopy, it is difficult to explore the entire bowel and hidden areas, such as behind the liver and the spleen, particularly in the presence of intra-abdominal adhesions [8].

To predict resectability, tumor biomarker CA-125 was assessed by different approaches. In the literature, the approaches are mostly based on the level of a single time point, such as cut-off levels of CA-125 at baseline [12], at the nadir during NAC [13] or normalization after NAC [14]. These approaches have major limitations due to the use of a unique level of CA-125 to characterize the complex kinetics of the biomarker without integrating the effect of chemotherapy and the timeframe [15]. Moreover, these simple approaches do not consider intra- and inter-individual variabilities related to patient and tumor features [15,16].

The CA-125 level changes over time during NAC and depends on tumor burden changes [17]. Modeling the CA-125 evolution during NAC considering clinical and oncological parameters may be an interesting strategy [15]. The joint model (JM) is of growing interest in the literature. The JM combines a Linear Mixed Effect (LME) sub-model with a Cox sub-model. The LME sub-model assesses longitudinal data of evolutive levels of biomarkers over time. The Cox sub-model assesses predictive time-to-event covariates [18]. The JM is used for dynamic prediction, where the prediction is updated based on the information that is available at a given time during follow-up [19]. Previously, a JM of longitudinal CA-125 was assessed as a reliable predictive model for overall and free disease survivals [20,21,22]. To our knowledge, no JM of CA-125 levels during NAC was developed in the literature to predict complete resectability of AOC.

This study aimed to develop a JM using patient and tumor features combined with CA-125 kinetics during NAC to predict complete resectability in patients with AOC.

## 2. Materials and Methods

### 2.1. Study Design and Population

We carried out a retrospective review of medical records from patients treated at Godinot Institute between January 2010 and December 2019. Only patients with International Federation of Gynecology and Obstetrics (FIGO) IIIc and IV [23] epithelial ovarian cancer who underwent NAC were included. Other ovarian cancer histology was not included. To build an evolutive model of CA-125 kinetics, patients with at least two CA-125 measurements during NAC were included.

### 2.2. Ethical Considerations

The study was approved by the French national data protection authority (CNIL-MR003 N° 2218430). The patients’ records were anonymized prior to analysis. A database was created in accordance with the reference methodology MR004 of the National Commission of Liberties and Informatics (n°2206749, 13 September 2018). A non-opposition form was sent to each living patient included in the study. As per French regulations, no additional ethical committee review was required.

### 2.3. Data Collection

The collected data were: age at diagnosis, body mass index (BMI), performance status (PS), tumor grade, peritoneal cancer index (PCI) [24], intra-abdominal involved organs, presence of extra-peritoneal metastases, chemotherapy regimens, biotherapy regimens, number of cycles, and number and volume of paracenteses. Paracenteses was performed during NAC in patients with ascites evaluated at grade 3 (large volume or symptomatic ascites) [25]. To avoid interlaboratory variability, we collected CA-125 levels that were provided by the same laboratory.

### 2.4. Management of Patients

The NAC was a platinum-based doublet with carboplatin and paclitaxel [26]. For patients in poor PS, a single-agent carboplatin was administered. Due to the side effects of chemotherapy in some patients, subsequent cycles of chemotherapy were postponed. Therefore, for these patients, the four cycles of chemotherapy were completed beyond the usual 12 weeks.

An explorative laparoscopy was performed to evaluate whether complete cytoreduction was feasible. A decision for complete resectability was made by skilled oncology gynecology surgeons in a multidisciplinary meeting according to either laparoscopic or imaging (CT scan) evaluations. Complete cytoreduction consists of the resection of all visible macroscopic nodules with no residual tumor in the peritoneal cavity. Non resectability criteria were: the tumor invasion of the mesentery root, hepatic hilum, pancreas and duodenum tract, a large segment of the bowel, and unresectable extra-abdominal metastases [27].

### 2.5. Statistical Analysis

#### 2.5.1. Fit a Linear Mixed-Effects (LME) Sub-Model

An LME sub-model was fitted to describe the evolution of CA-125 over time [28]. Repeated CA-125 levels were recorded from CA-125 measured within one month before the first NAC cycle up to CA-125 measured after the third or fourth NAC cycle. In order to record homogeneous patients’ data, CA-125 measured after four cycles of NAC were not included. Additionally, the LME sub-model included the covariates that may influence the evolution of CA-125 levels. The logarithmic transformation of the square-root of CA-125 was applied to respect the residual’s homoscedasticity and linear relationship over time [21].

#### 2.5.2. Fit a Cox Sub-Model

A Cox proportional hazards sub-model was used to screen covariates that may be associated with resectability, such as age, PS, FIGO classification, tumor grade, PCI, location of peritoneal carcinomatosis (PC) at baseline, presence of ascites, CA-125 level at baseline, and chemotherapy regimens. The subjects were censored at the time when the operability decision was made (resectable: yes vs. no).

#### 2.5.3. Fit the Joint Model

The JM linked the best LME sub-model with the Cox sub-model including significant covariates in univariate analysis. A Bayesian approach was used where inference was based on the posterior of the model.

Validation of the final model was assessed by 5-fold cross validation. The cohort was randomly divided into five subgroups of equal size. The analysis was repeated five times, using one subgroup as the test subgroup and the others as the training subgroup each time. The parameter estimates of the JM were derived from the training subgroup and applied to the test subgroup. The area under the receiver operating characteristic curve (AUC) was calculated to assess the discriminative capability of the model. The accuracy was assessed by using predictive error (PE) [29,30].

As an example, we applied the fitted JM to 4 patients and predicted their future resectability.

## 3. Results

### 3.1. Population Characteristics

From January 2010 to December 2019, 156 patients were referred to our institute for AOC. Among them, 114 were initially unresectable. In this study, 77 patients with AOC were included. The remaining patients were not included for the reasons shown in the flowchart in Figure 1.

The overall population characteristics along with the univariate analysis, according to resectability (Complete ICRS: yes/no), are presented in Table 1.

Complete ICRS was achieved in 40 patients (52%). For 37 patients (48%), complete resection was not feasible mainly because of tumor spread involving a large part of the bowel, the hepatic or portal veins, the iliac vessels, or because of unresectable extra-abdominal metastases. The median PCI at diagnosis, calculated on diagnostic laparoscopy, was not significantly different between the resectable and unresectable patients. A PC located on the hepatic hilum or on the stomach was significantly associated with non-resectability (*p* = 0.009). Ascites was present in 68 patients (88.3%) at diagnosis. During NAC, for 12 patients (15.6%), iterative paracenteses of at least 1.5 L was necessary because of symptomatic ascites.

The median number of NAC cycles was 6 (interquartile (IQR) range, 4 to 6). Thirty-one patients (40.2%) had three or four cycles. Other patients had five to nine cycles. For these latter patients, the decision for additional cycles was made because of insufficient tumor response on CT scan or on interval laparoscopy. Among them, 24 patients (31%) became eligible for ICRS.

### 3.2. Factors Related to Resectability

Fitting multivariate Cox proportional hazards model, a poor PS (HR = 0.57, CI 95% [0.34–0.95], *p* = 0.03), and the presence of PC located on the hepatic hilum or on the stomach (HR = 0.39, CI 95% [0.19–0.80], *p* = 0.01) were covariates associated with non resectability.

### 3.3. CA-125 Kinetics during NAC

A total of 389 CA-125 values were recorded with a median of 5 [2–6] samples per patient. The median CA-125 at diagnosis was 926 IU/mL [IQR 23–27,350]. CA-125 at diagnosis was not significantly associated with resectability. The CA-125 level after NAC was significantly different between the resectable and unresectable patients (*p* < 0.0001). Figure 2 illustrates the evolution of CA-125 for each patient according to ICRS feasibility. The LME sub-model described the evolution of CA-125 from diagnosis to the completion of NAC with covariates influencing the levels of CA-125. CA-125 decreased as the number of NAC cycles increased (HR = 0.89, *p* = 0.05) and as paracenteses were necessary (HR = 0.76, *p* = 0.006).

### 3.4. Joint Model (JM) of Longitudinal CA-125 and Tumor Resectability

The JM linked CA-125 kinetics adjusted to the number of NAC cycles and the number of paracenteses in the LME sub-model, with covariates associated with resectability selected from the Cox sub-model (worst PS and the presence of PC in the epigastrium region). The accuracy of this JM was evaluated by a cross validation method. The average of accuracy prediction was 88% ranging from 82 to 100%. The average of predictive errors was 2.24% ranging from 0 to 2.32%.

Table 2 depicts the results of previous studies that focused on the association between the cut-off level of CA-125 (75 and 20 IU/mL) and the resectability of AOC after NAC. These results were tested on our population study. In our population, CA-125 less than 75 IU/mL after the third cycle was significantly associated with resectability (OR = 3.56, CI 95% [1.57–8.08], *p* = 0.002), whereas CA-125 < 20 IU/mL before ICRS was not significantly associated to resectability (*p* = 0.25).

To illustrate how the JM works, the probability of resectability was plotted for four patients. To dynamically estimate the probability of resectability using the fitted JM, we recorded the available baseline patient characteristics and CA-125 levels from diagnosis to the end of NAC. Figure 3 represents the dynamic prediction of the event probability for patients A, B, C, and D respectively. On the left of the vertical dotted line is the longitudinal trajectory of the CA-125 levels. On the right is the probability of resectability. Specifically, patient A had a PS = 1 with no PC on the epigastrium region at diagnosis. No paracentesis was performed during NAC. This patient underwent 6 courses of chemotherapy of carboplatin-paclitaxel. Her CA-125 levels over time were 3328, 830, 121, 37, 19, and 16 IU/mL measured at times 0, 3, 6, 9, 12, and 15 weeks, respectively. Before NAC, the model predicted her complete ICRS probability under 60%. After NAC, the model predicted her complete ICRS probability close to 80%. Indeed, patient A had a complete ICRS. Patient B was estimated with a PS = 0 and the presence of PC located on the epigastrium region at diagnosis. Two paracenteses were performed during NAC. Patient B underwent 4 courses of chemotherapy of carboplatin-paclitaxel. The CA-125 levels during NAC were 748, 706, and 649 IU/mL at 0, 3, and 6 weeks, respectively. Before NAC, the probability of complete ICRS considering the patient characteristics and first CA-125 was up to 20%. As new CA-125 levels were recorded, the probability of complete ICRS after NAC was 36%. For patient B, complete ICRS was considered as not feasible after explorative laparoscopy. Patient C was estimated with a PS = 1 and the presence of PC located on the epigastrium region at diagnosis. No paracentesis was performed during NAC. This patient underwent 6 courses of chemotherapy of carboplatin-paclitaxel. The CA-125 levels during NAC were 679, 91, 48, and 44 IU/mL at 0, 9, 14, and 17 weeks, respectively. The probability of complete resection was adjusted as the CA-125 levels were recorded from 41% before NAC to 75% after NAC. Patient C had a complete ICRS. Patient D was estimated with a PS = 1 and the presence of PC located on the epigastrium region at diagnosis. One paracentesis was performed during NAC. This patient underwent 6 courses of chemotherapy of carboplatin-paclitaxel. The CA-125 levels during NAC were 5192, 2057, 344, 93, 37, and 19 IU/L at 0, 3, 6, 9, 12, and 17 weeks, respectively. Before NAC, the JM predicted her complete ICRS probability up to 45%. After NAC, the model predicted her complete ICRS probability close to 60% after NAC completion. For patient D, complete ICRS was not feasible.

## 4. Discussion

In this study, we developed a dynamic and individual accurate predictive model of complete ICRS feasibility in patients with AOC. As part of the JM, an LME sub-model included the number of iterative paracentesis and the number of NAC cycles as covariates affecting the trajectory of CA-125 levels. This allowed a better modeling of the CA-125 kinetics. As predictive covariates, poor PS and the presence of PC in the epigastrium region were included in the Cox sub-model. This JM, with an accuracy > 85% and a predictive error < 3%, showed the reliability of the model to predict the resectability of the AOC after NAC.

The kinetics of CA-125 levels is correlated to tumor size evolution during NAC, suggesting a perspective to predict resectability [17,34]. Indeed, as a result of the NAC administration, CA-125 levels decrease. Also, the CA-125 level is correlated to the presence and the amount of ascites [35,36].

A poor PS is a well-known risk factor of non-resectability [37]. In our population study, the presence of PC in the epigastrium region (omentum lesser sac, the stomach, and hepatic pedicle) was prohibitive for resectability. This may be explained by the fact that in AOC, a posterior pelvectomy including a left colectomy is often necessary, and adding a gastrectomy when the gastric pedicle is involved is not recommended [38].

A CA-125 kinetic modeling approach considering inter- and intra-individual variability is more reliable than a selected time points level [16]. In clinical practice, surgeons and medical oncologists intuitively predict the probability of tumor response to NAC on the basis of the patients’ state (such as age, PS, etc.) and disease state (disease stage and PCI and CA-125 levels after NAC). The JM includes baseline patient and disease features combined with the evolution of the biomarker over time considering factors that may influence the biomarker levels. Therefore, the JM may be a reliable tool to help physicians to predict tumor resectability.

Some studies have shown that a JM with longitudinal CA-125 levels is an accurate model to predict survival [20,21,22]. In these studies, the estimation of patients’ survival probability was improved by using significant medical data, which can be accumulated during follow-up. Another interesting ability of the JM is the dynamic prediction of the probability of the event over time as new CA-125 levels are recorded [20]. Moreover, the accuracy of predictions increases with time [30]. The JM is a personalized model because it considers demographic and clinical characteristics of the patient and compares CA-125 levels to their own reference and not to a median or average.

Studies that have determined a cut-off level for CA-125 below which resectability may be possible, reported limited predictive capability as their AUC remains below 80% [39,40]. These findings were poorly reproducible on other cohorts of patients [41]. These cut-off levels were tested on our population study and their predictive capability were less than 75% (Table 2).

A semi-mechanistic model based on kinetic-pharmacodynamics KELIM (modeled CA-125 elimination rate constant) approach was developed to fit the CA-125 levels during NAC [17,42]. The authors derived a model based on tumor biomarker elimination parameter (KELIM) as predictive of completeness of ICRS after NAC. A limitation could be that patient baseline characteristics are not included in the model and therefore may be not adapted for all clinical situations, such as advanced age or poor PS.

The JM developed in our series may be used as a complement tool of imaging and exploratory laparoscopy in the prediction of complete resectability of AOC. Furthermore, one of the perspectives could be that oncologists may detect early non-responder patients to NAC and eventually readjust chemotherapy regimens.

This study has some limitations. First, this is a retrospective single centre study. Therefore, the results must be validated in another cohort of patients. Second, in our population, 60% of patients with initially unresectable AOC received more than four cycles of NAC. Additional cycles of NAC were suggested for patients with a partial but insufficient tumor response after four cycles. The goal of additional cycles of NAC was to achieve complete cytoreduction (R0) while reducing surgical complexity and allowing organ preservation. Moreover, resectability criteria differ according to various centres and surgeons’ experiences. In our centre, the decision of resectability was made in a multidisciplinary gynecological oncology meeting with two oncological skilled surgeons on the basis of worldwide consensual criteria. In patients who received more than four cycles of NAC followed by a complete ICRS (*n* = 25), the majority of patients (*n* = 21) had three or four cycles of adjuvant chemotherapy with carboplatin and paclitaxel within 12 weeks with bevacizumab. The other four patients received carboplatin monotherapy (due to paclitaxel-induced neuropathy).

## 5. Conclusions

The JM applied to CA-125 kinetics during NAC is helpful for clinicians to predict complete cytoreduction in patients with AOC and to select patients for ICRS.

## Figures and Tables

**Figure 1 cancers-15-00231-f001:**
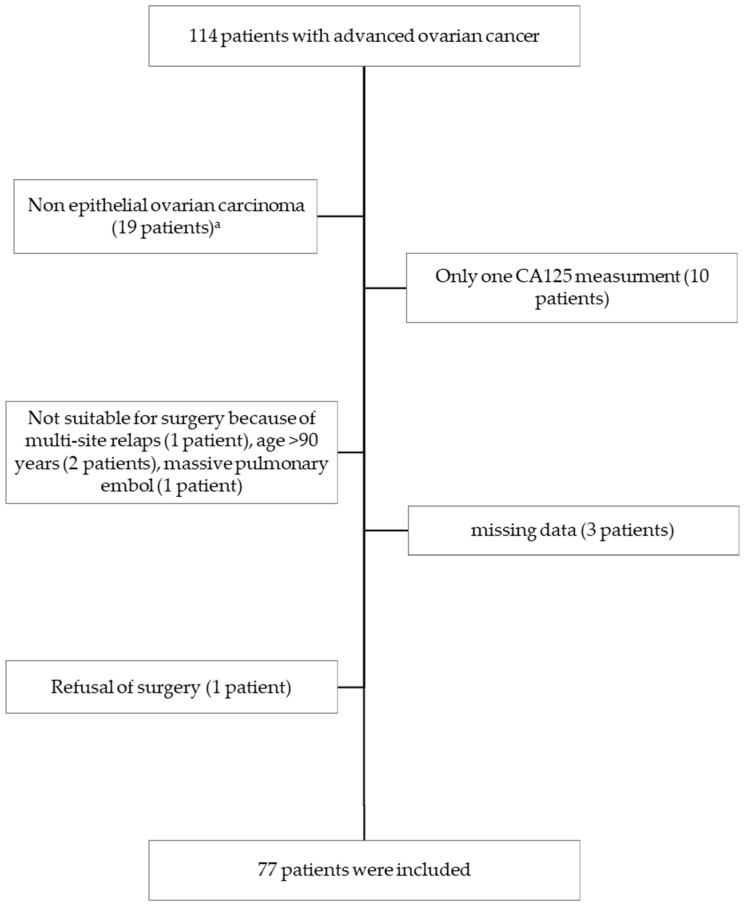
Flow Chart. ^a^ 5 borderline, 4 mucinous, 3 carcinosarcoma, 2 endometrioid, 1 Sertoli cell, 1 granulosa tumor, 1 independent cell, 1 clear cell, and 1 mixed Mullerian.

**Figure 2 cancers-15-00231-f002:**
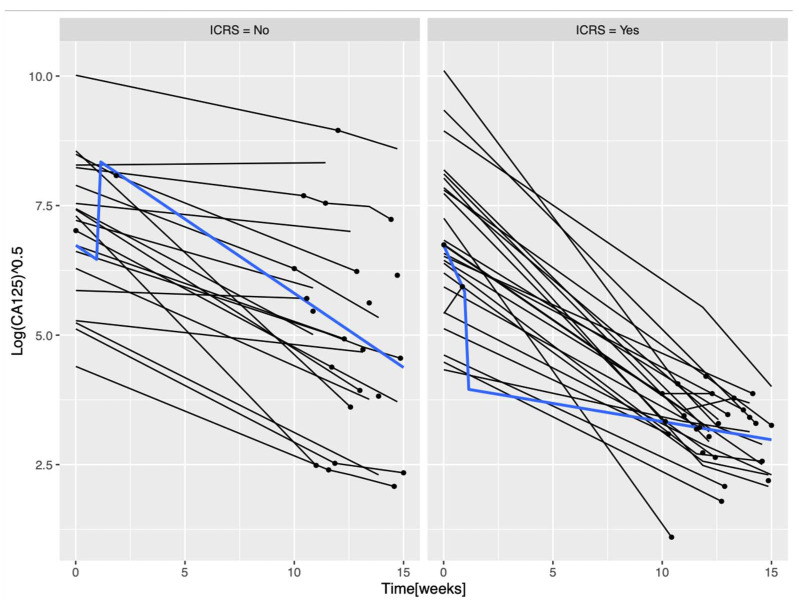
Plot longitudinal CA-125 levels. Individual evolution in time of the CA-125 values, separately for resectable (IRCS = yes) and non-resectable (ICRS = no) patients. The blue line represents the median value of CA-125. Abbreviations: ICRS: interval cytoreductive surgery.

**Figure 3 cancers-15-00231-f003:**
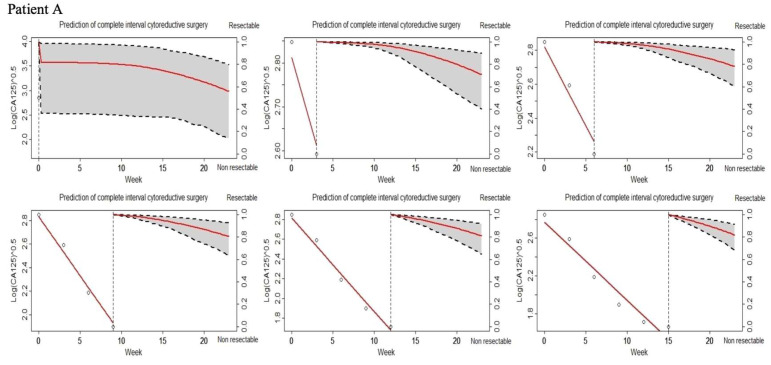
Application of a joint model for dynamic prediction of the probability of complete interval cytoreductive surgery (ICRS) for patients A, B, C, and D. The longitudinal CA-125 values are represented on the left of the vertical dotted line. On the right of the vertical dotted line is the probability of the feasibility of a complete ICRS. The probabilities of complete ICRS were updated as new CA-125 values were recorded. Patient A with a PS = 1 and without PC located on the epigastrium region at diagnosis. No paracentesis was performed during NAC. This patient underwent 6 courses of NAC. Before NAC, the probability of complete ICRS considering patient characteristics and first CA-125 was under 60%. As new CA-125 values were recorded during NAC, the prognosis was updated to a probability of complete ICRS of 75%. A complete ICRS was performed. Patient B with a PS = 0 and the presence of PC located on the epigastrium region at diagnosis. Two paracenteses were performed during NAC. This patient underwent 4 courses of NAC. Initial probability of complete ICRS considering patient characteristics and first CA-125 was up to 20%. As new CA-125 values were entered until the end of NAC, the prognosis was updated to a probability of complete ICRS of 36%. For patient B, complete ICRS was not feasible. Patient C was estimated with a PS = 1 and the presence of PC located on the epigastrium region at diagnosis. No paracentesis was performed during NAC. This patient underwent 6 courses of NAC. Initial probability of complete ICRS considering patient characteristics and first CA-125 was close to 40%. As new CA-125 values were entered until the end of NAC, the prognosis was updated to a probability of complete ICRS of close to 80%. For patient C, complete ICRS was performed. Patient D was estimated with a PS = 1 and the presence of PC located on the epigastrium region at diagnosis. One paracentesis was performed during NAC. This patient underwent 6 courses of NAC. Initial probability of complete ICRS considering patient characteristics and first CA-125 was close to 45%. As new CA-125 values were entered until the end of NAC, the prognosis was updated to a probability of complete ICRS of 60%. For patient D, complete ICRS was not feasible.

**Table 1 cancers-15-00231-t001:** Demographic, clinical, and biological characteristics.

Characteristics	Overall Population (%)	Complete Interval Cytoreductive Surgery (%)		*p*
		No *n* = 37 (48%)	Yes *n* = 40 (52%)	0.20
Age > 65 (years)	40 (52)	22 (55)	18 (45)	0.46
BMI (kg/m^2^)	24.2 [21.5–29.7]	25.1 [21.8–29.7]	22.6 [21.3–28.8]	0.36
PS				
0	44 (57.1)	18 (48.6)	26 (65)	
1	26 (33.8)	14 (37.8)	12 (30)	
2	6 (7.8)	4 (10.8)	2 (5)	
3	1 (1.3)	1 (2.7)	0 (0)	
Age at menopause (years), Median [IQR]	50 [50–53]	50.0 [50.0–53.5]	50.0 [46.8–53.0]	0.53
Histology sub-type				0.42
Serous	71 (92.2)	33 (89.2)	38 (95)	
Mixed or Undifferentiated	6 (7.8)	4 (10.8)	2 (5)	
Tumor grade				<0.01
I	3 (4.6)	3 (9.7)	0(0)	
II	49 (75.4)	17 (54.8)	32 (94.1)	
III	13 (20)	11 (35.5)	2 (5.9)	
FIGO stage				0.43
IIIc	66 (85.7)	30 (81.1)	36 (90)	
IVa	5 (6.5)	4 (10.8)	1 (2.5)	
IVb	6 (7.8)	3 (8.1)	3 (7.5)	
Extra-abdominal metastases				0.58
Extra-abdominal Lymph nodes	4 (5.2)	2 (5.4)	2 (5)	
Liver	2 (2.6)	1 (2.7)	1 (2.5)	
Pleural	5 (6.5)	4 (10.8)	1 (2.5)	
CA-125 Pre-NAC (IU/mL), median [range]	926 [23–27,350]	1115 [23–27,350]	848 [39–24,465]	0.75
Ascites at diagnosis	68 (88.3)	34 (91.9)	34 (85)	0.48
Number of ascites punctures				0.06
1	3 (3.9)	1 (2.7)	2 (5)	
2	7 (9.1)	6 (16.2)	1 (2.5)	
4	1 (1.3)	1 (2.7)	0 (0)	
6	1 (1.3)	1 (2.7)	0 (0)	
PCI at diagnosis, median [range]	21 [18–29]	24 [18–29]	20 [17–23]	0.11
PC location details at diagnosis				
Douglas	67 (87)	34 (91.9)	33 (82.5)	0.31
Hepatic hilum or stomach	30 (39)	20 (54.1)	10 (25)	<0.01
Small intestine	52 (67.5)	28 (75.7)	24 (60)	0.14
Large intestine	49 (64.5)	23 (63.9)	26 (65)	0.92
Right diaphragmatic dome	58 (76.3)	31 (83.8)	27 (69.2)	0.14
Left diaphragmatic dome	50 (65.8)	28 (75.7)	22 (56.4)	0.08
NAC regimen				1.00
Carboplatin-paclitaxel	73 (94.8)	35 (94.6)	38 (95)	
Carboplatin	4 (5.2)	2 (5.4)	2 (5)	
Biotherapy				0.22
Bevacizumab	4 (5.2)	3 (8.1)	1 (2.5)	
Pembrolizumab	8 (10.4)	3 (8.1)	5 (12.5)	
TSR 042 or placebo	3 (3.9)	3 (8.1)	0 (0)	
Cycles of NAC median [range]	6 [3–9]	6 [3–9]	6 [3–7]	0.95
CA-125 < 75 after 3rd NAC	46 (59.7)	13 (35.1)	33 (82.5)	<0.01
Post-NAC CA-125 (IU/mL), median [range]	35 [16–144]	85 [35–402]	22 [13–38]	<0.01

Abbreviations: IQR: interquartile range; BMI: Body Mass Index; PS: Performance status; FIGO: International Federation of Gynecology and Obstetrics; NAC: Neo adjuvant chemotherapy; PCI: Peritoneal Cancer Index; PC: Peritoneal carcinomatosis.

**Table 2 cancers-15-00231-t002:** Literature review of preoperative CA-125 levels after NAC and prediction of optimal/complete cytoreductive surgery of AOC.

Author (Year)	Type of Study	Endpoint	Number of Patients	CA-125 Cut-Off (IU/mL)	AUC (%)(CI)	AUC in Our Study Using Literature Cut-Off
**Shen et al., 2016** [31]	Retrospective	ICRS	43	58.58	66 [50–83]	71 [62–80]
**Furukawa et al., 2013** [32]	Retrospective	Complete and non-complete ICRS	75	20	NA	63 [53–73]
**Pelissier et al., 2013** [33]	Retrospective	Optimal ICRS	148	75 after the 3rd NAC	73 [62–82]	70 [59–80]
**Our study**	Retrospective	Complete ICRS	77	-	88 [82–100]	

Abbreviations: Se: Sensibility; Sp: Specificity; AUC: area under the curve; ICRS: Complete interval cytoreductive surgery; NA: not available; NAC: Neo adjuvant chemotherapy; PCI: Peritoneal Cancer Index; PC: Peritoneal carcinomatosis.

## Data Availability

The data presented in this study are available on request from the corresponding author.

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
