# Peer review of "Dynamic Prediction of Resectability for Patients with Advanced Ovarian Cancer Undergoing Neo-Adjuvant Chemotherapy: Application of Joint Model for Longitudinal CA-125 Levels"

_cancers, 2022, doi:10.3390/cancers15010231_

Round 1

Reviewer 1 Report

Dear Editor, 

I would like to congratulate Amroun et al. on such an original work. In my opinion, new statistical tools as JM may help us solve daily problems as complete resectability in AOC. 

Nevertheless, I'd like to raise attention on some specific points of the manuscript: 

- Methods section, paragrapgh 2.4: I understand that every included patient were treated by NAC + ICRS. In fact, in the results section (3.1), it is stated that 114 patients were adressed to the authors hospital for AOC, being included 77 (excluding the rest of the patients for several reasons). As it is standard practice to perform primary cytoreductive surgery when feasible (and this practice stands as a quality recommendation in the ESGO guidelines), which were the reasons not to perform PDS in any of the AOC admitted patients? Is it a standard practice at this institution to perform NAC in every advanced AOC patient at diagnosis, irrespective of their initial PCI? Is HIPEC performed as standard practice in this setting? Please clarify these important aspects of the clinical management of AOC patients. 

- Table 1: I think that there is a typo in the title of the Table, as it stands as "Interval cytoreductive surgery (%)", and it seems (as can be understood by the text) that it should refer to Complete cytoreductive surgery. 

- In Table 1, PCI at diagnosis is stated. Was this PCI calculated by imaging techniques or diagnostic laparoscopy? I think that it would be interesting to assess PCI at the moment of the interval laparoscopy too, as it would be an important prognostic tool to know if complete cytoreduction can be achieved. 

- Cuantitative variables are expressed as median, and IQR or range?

- How could the authors explain that only 40% of the patients received 3-4 cycles, while 60% received more than 4 cycles as NAC? As it is not a standard practice while managing NAC in AOC patients, please explain. It cannot be considered ICRS when 5-9 cycles of NAC are administered before surgery. In my opinion, there are some non respondent patients that would need some extra cycles of CT before attempting surgery, but it is not standard that this group encompasses the majority of NAC treated patients. Also, I would like to know if, after 5-9 cycles and complete surgery, more CT cycles were administered and under which CT regime. 

Author Response

Thanks to the Reviewer for appreciating our work and for the time she/he spent in revising it.

Point 1: - Methods section, paragraph 2.4: I understand that every included patient were treated by NAC + ICRS. In fact, in the results section (3.1), it is stated that 114 patients were addressed to the authors hospital for AOC, being included 77 (excluding the rest of the patients for several reasons). As it is standard practice to perform primary cytoreductive surgery when feasible (and this practice stands as a quality recommendation in the ESGO guidelines), which were the reasons not to perform PDS in any of the AOC admitted patients? Is it a standard practice at this institution to perform NAC in every advanced AOC patient at diagnosis, irrespective of their initial PCI? Is HIPEC performed as standard practice in this setting? Please clarify these important aspects of the clinical management of AOC patients. 

Response 1: In fact 156 patients wit AOC were adressed to our institution. Forty-two of them were initially resectable and underwent surgery. Whereas 114 patients were not resectable at diagnosis. From these patients 77 were included in our study. The corrections are reported in the sub-section “ 3.1. Population characteristics” of the “Results” section.

Point 2: - Table 1: I think that there is a typo in the title of the Table, as it stands as "Interval cytoreductive surgery (%)", and it seems (as can be understood by the text) that it should refer to Complete cytoreductive surgery.

Response 2: Correction made in table 1

Point 3: - In Table 1, PCI at diagnosis is stated. Was this PCI calculated by imaging techniques or diagnostic laparoscopy? I think that it would be interesting to assess PCI at the moment of the interval laparoscopy too, as it would be an important prognostic tool to know if complete cytoreduction can be achieved. 

Response 3: PCI at diagnosis was calculated by diagnostic laparoscopy. This has been clarified in the “3.1. Population characteristics” sub-section of the “Results” section.

Point 4: - Quantitative variables are expressed as median, and IQR or range?

Response 4: Age at menopause was expressed as median and IQR. CA-125 Pre-NAC, PCI at diagnosis, Cycles of NAC and Post-NAC CA-125 were expressed as median and range. A rectification is made for ”Cycles of NAC” in the table 1 to be expressed as median and range.

Point 5: - How could the authors explain that only 40% of the patients received 3-4 cycles, while 60% received more than 4 cycles as NAC? As it is not a standard practice while managing NAC in AOC patients, please explain. It cannot be considered ICRS when 5-9 cycles of NAC are administered before surgery. In my opinion, there are some non respondent patients that would need some extra cycles of CT before attempting surgery, but it is not standard that this group encompasses the majority of NAC treated patients. Also, I would like to know if, after 5-9 cycles and complete surgery, more CT cycles were administered and under which CT regime.

Response 5: The reviewer pointed that 60% (n= 46) of patients with initially unresectable AOC received more than 4 cycles NAC, within a majority (n= 44) received 5 to 7 cyles NAC. Among these patients, complete ICRS was possible in 25 patients. Two patients received 9 cycles low dose NAC. There was no significant difference in the number of NAC between patients in whom complete ICRS was possible versus not. The purpose of additionnal cycles of NAC was to achieve a complete (R0) cytoreduction allowing organ preservation and reducing complexity of surgery. Additionnal NAC cycles was proposed to patients in whom a tumor partial response, but insufficient, was observed after 4 cycles. Scince the level of CA-125 is influenced by the number of chemotherapy cycles, this was included in the LME sub-model of our analysis.

In patients with ≥ 5 cycles NAC followed by a complete ICRS (n= 25), for 21 patients, 3 or 4 cycles adjuvant chemotherapy was conducted with carboplatin and paclitaxel within 12 with bevacizumab according to multidisciplinary meeting decisions. For the four other patients, four received carboplatine monotherapy (du to paclitexel induced neupathy).

If the reviewer wishes, this answer can be added to the discussion.

Reviewer 2 Report

I would like to congratulate the autohrs for the initiative. 

The manuscript describes an alternative to correctly predict surgical resecability in ovarian cancer patients submitted to NACT. 

It is an important study that reinforces the role of CA-125 in the follow-up of NACT, not only to adress the response to CT but also to avoid unnecessary operations. 

Author Response

Thanks to the Reviewer for appreciating our work.

Reviewer 3 Report

Ovarian cancer presents at an advanced stage when the cancer has already metastasized. Cytoreductive surgery followed by chemotherapy is the standard mode of treatment when cytoreduction is possible. However, in patients when cytoreduction is not possible, neoadjuvant chemotherapy is used.  As the tumor burden changes, CA-125 levels also change.  Here, the authors have developed a Joint Model (JM) as a statistical tool to predict the overall and disease-free survival of patients receiving neoadjuvant chemotherapy.  This tool uses patient’s and tumor characteristics and kinetics of CA-125 during neo-adjuvant chemotherapy to predict if the tumors are resectable or not. This is a very useful model especially for patients receiving neoadjuvant chemotherapy and could help surgeons adjust their chemotherapy regimens based on tumor growth in patients.

Author Response

On behalf of all authors, we would like to thanks the Reviewer for appreciating our work.